# High-resolution mapping demonstrates inhibition of DNA excision repair by transcription factors

**Mingrui Duan[1], Smitha Sivapragasam[2], Jacob S Antony[2], Jenna Ulibarri[1], John M Hinz[2], Gregory MK Poon[3], John J Wyrick[2,4]\*, Peng Mao[1]\***

[1]Department of Internal Medicine, University of New Mexico Comprehensive Cancer Center, University of New Mexico, Albuquerque, United States; [2]School of Molecular Biosciences, Washington State University, Pullman, United States; [3]Department of Chemistry, Georgia State University, Atlanta, United States; [4]Center for Reproductive Biology, Washington State University, Pullman, United States

**\*For correspondence:**
jwyrick@wsu.edu (JJW);
pmao@salud.unm.edu (PM)

**Competing interest:** The authors declare that no competing interests exist.

**Abstract** DNA base damage arises frequently in living cells and needs to be removed by base excision repair (BER) to prevent mutagenesis and genome instability. Both the formation and repair of base damage occur in chromatin and are conceivably affected by DNA-binding proteins such as transcription factors (TFs). However, to what extent TF binding affects base damage distribution and BER in cells is unclear. Here, we used a genome-wide damage mapping method, *N*-methylpurine-sequencing (NMP-seq), and characterized alkylation damage distribution and BER at TF binding sites in yeast cells treated with the alkylating agent methyl methanesulfonate (MMS). Our data show that alkylation damage formation was mainly suppressed at the binding sites of yeast TFs ARS binding factor 1 (Abf1) and rDNA enhancer binding protein 1 (Reb1), but individual hotspots with elevated damage levels were also found. Additionally, Abf1 and Reb1 binding strongly inhibits BER in vivo and in vitro, causing slow repair both within the core motif and its adjacent DNA. Repair of ultraviolet (UV) damage by nucleotide excision repair (NER) was also inhibited by TF binding. Interestingly, TF binding inhibits a larger DNA region for NER relative to BER. The observed effects are caused by the TF–DNA interaction, because damage formation and BER can be restored by depletion of Abf1 or Reb1 protein from the nucleus. Thus, our data reveal that TF binding significantly modulates alkylation base damage formation and inhibits repair by the BER pathway. The interplay between base damage formation and BER may play an important role in affecting mutation frequency in gene regulatory regions.

## Editor's evaluation

This manuscript will be of interest to researchers interested in DNA repair and transcriptional regulation. The authors provide a series of well-executed and designed high-resolution sequencing data demonstrating that transcription factor (TF) binding perturbs alkylation base damage formation as well as inhibits its repair via base excision repair (BER) at TF binding sites. Moreover, they demonstrate differences between nucleotide excision repair and BER at TF binding sites that are consistent with the different repair mechanism of these two pathways. These results should have an important and timely impact on the field. The revision addresses all points of the review and has significantly strengthened the manuscript and presentation.

## Introduction

DNA in living cells is exposed to an array of genotoxic agents, both endogenous and exogenous. Alkylating agents comprise a large number of reactive chemicals present in cells and in the environment (*Fu et al., 2012*), which can react with the nitrogen and oxygen atoms of DNA bases to induce formation of alkylation damage. Some alkylation damage is cytotoxic and mutagenic (*Kondo et al., 2010*), and thus poses threats to cell growth and genome stability. On the other hand, the cytotoxicity of DNA alkylation is utilized in chemotherapy. Alkylating agents such as temozolomide (TMZ) are used for the treatment of glioblastoma and other cancers (*Fu et al., 2012*; *Newlands ES, Stevens MFG, Wedge SR, Wheelhouse RT, Brock C, 1997*). Therefore, studies of alkylation damage and its repair are relevant for both cancer prevention and therapy.

The most common alkylation lesions are *N*-methylpurines (NMPs), including 7-methylguanine (7meG) and, to a lesser extent, 3-methyladenine (3meA) (*Kondo et al., 2010*). Although 7meG is not genotoxic by itself, it is prone to spontaneous depurination to form a mutagenic apurinic (AP) site (*Fu et al., 2012*). 7meG can also form deleterious DNA–protein crosslinks with the lysine-rich histone tails (*Yang et al., 2018*). The 3meA damage is even more harmful than 7meG, as 3meA lesions block DNA polymerases and affect DNA replication (*Plosky et al., 2008*). Hence, NMP lesions need to be repaired in a timely manner to avoid detrimental outcomes such as cell death or mutations. The primary repair pathway for NMPs is base excision repair (BER), which is initiated by alkyladenine-DNA glycosylase (AAG; also known as MPG and ANPG) in human cells or its yeast ortholog Mag1 (*O'Connor and Laval, 1991*). During BER, AAG/Mag1 removes the alkylated base and generates an AP site, which is then cleaved by the apurinic/apyrimidinic endonuclease (APE1) (*Mol et al., 2000*). Subsequently, DNA polymerase and ligase are recruited to the nick to conduct repair synthesis and ligation, respectively (*Krokan and Bjørås, 2013*).

Transcription factors (TFs) are key proteins that regulate gene expression. Many TFs bind to DNA in a sequence-specific manner to direct transcription initiation to target promoters (*Jolma et al., 2013*). While TFs mainly function in transcriptional regulation, their binding to DNA can affect DNA damage formation and repair (*Mao and Wyrick, 2019*). To this end, several TF proteins have been shown to modulate formation of ultraviolet (UV) light-induced photolesions (*Frigola et al., 2021*; *Hu et al., 2017*; *Mao et al., 2018*) and inhibit nucleotide excision repair (NER) (*Conconi et al., 1999*; *Sabarinathan et al., 2016*). The altered UV damage formation and suppressed NER are believed to cause increased mutation rates at TF binding sites in skin cancers (*Frigola et al., 2021*; *Mao et al., 2018*; *Sabarinathan et al., 2016*). Previous studies have also found that mutation rates are significantly increased at TF binding sites in non-UV exposed tumors (*Kaiser et al., 2016*; *Melton et al., 2015*), such as gastric and prostate cancers (*Guo et al., 2018*; *Morova et al., 2020*). However, what causes the high mutation rates in non-UV exposed cancers remains elusive. Since base damage (e.g. oxidative, alkylation, uracil, and so on) caused by endogenous and exogenous damaging sources is prevalently associated with cancer mutations (*Tubbs and Nussenzweig, 2017*; *Wallace et al., 2012*), a potential mechanism for mutation elevation in non-UV exposed tumors is increased base damage formation and/or suppressed BER in TF-bound DNA. However, this hypothesis has not been tested and it is unclear to what extent TF binding affects base damage formation and BER.

Alkylation damage has been widely used as a model lesion for BER studies (*Fu et al., 2012*; *Li et al., 2015*). We previously developed an alkylation damage mapping method, *N*-methylpurine-sequencing (NMP-seq), to precisely map 7meG and 3meA lesions in cells treated with methyl methanesulfonate (MMS) (*Mao et al., 2017*). Here, we used NMP-seq to analyze alkylation damage formation and BER at the binding sites of ARS binding factor 1 (Abf1) and rDNA enhancer binding protein 1 (Reb1), two essential yeast TFs that have been extensively characterized. The genome-wide binding sites for Abf1 and Reb1 have been identified at near base-pair resolution (*Kasinathan et al., 2014*; *Rossi et al., 2021*) and the DNA-binding mechanisms were analyzed in previous studies (*Jaiswal et al., 2016*; *McBroom and Sadowski, 1994a*). Analysis of our NMP-seq data indicates that both damage formation and BER are affected by TF binding in yeast cells. We further show that Reb1 protein binding directly inhibits BER of alkylation damage in vitro. Collectively, these analyses uncover an important role for TF binding in modulating base damage formation and inhibiting BER.

## Results

### Abf1 and Reb1 modulate alkylation damage formation at their binding sites

NMP-seq is a sequencing method developed to map 7meG and 3meA lesions across the genome (*Mao et al., 2017*). This method employs BER enzymes AAG and APE1 to digest MMS-damaged DNA and create a nick at the NMP lesion site, which is then ligated to adaptor DNA for next-generation sequencing (*Figure 1—figure supplement 1*). As NMP lesion sites are precisely tagged by the adaptor DNA, sequencing with a primer complementary to the adaptor generates a genome-wide profile of NMP lesions at single-nucleotide resolution (*Mao et al., 2017*).

To determine how TF binding affects NMP lesion formation, we analyzed initial NMP lesions at Abf1 and Reb1 binding sites in yeast immediately after 10 min MMS treatment (i.e. no repair incubation). The ongoing BER during the period of MMS exposure may repair some of the damage and affect analysis of NMP formation. To minimize the effect of endogenous BER, we used a BER-deficient *mag1* deletion strain (i.e. *mag1Δ*) to profile the initial NMP distribution. We obtained a total of ~44 million sequencing reads in MMS-treated *mag1Δ* cells. The majority of the reads (~56%) were associated with G nucleotides (G reads), followed by A nucleotides (A reads) (*Figure 1—figure supplement 1*), consistent with the expected trend of 7meG and 3meA lesion formation after MMS treatment (*Friedberg et al., 2006*).

Since 7meG is the major class of lesion induced by MMS, we first characterized 7meG formation at Abf1 and Reb1 binding sites. To account for potential DNA sequence bias at the binding sites, we also mapped NMP damage in naked yeast genomic DNA, in which all proteins were removed and the purified DNA was damaged by incubating with MMS (*Figure 1—figure supplement 1*). Normalization of cellular G reads by the naked DNA G reads enables us to elucidate the modulation of 7meG formation by TF proteins. Importantly, we found that formation of 7meG was significantly inhibited at Abf1 and Reb1 binding sites relative to the flanking DNA (*Figure 1A, B*). Analysis of the average 7meG levels in 5 bp non-overlapping moving windows indicates that 7meG was reduced by up to 40 and 70% for Abf1 and Reb1 binding sites, respectively, and the difference is statistically significant (*Figure 1—figure supplement 2*). Furthermore, the extent of damage reduction was correlated with the level of TF occupancy, as Reb1 binding sites with low occupancy (occupancy <10) (*Kasinathan et al., 2014*) only slightly reduced 7meG formation (*Figure 1C*).

Damage formation was further analyzed in the TF core motif and its immediately adjacent DNA (20 bp on each side of the motif midpoint). This analysis shows that 7meG formation was significantly suppressed in the conserved regions of the motif sequences (*Figure 1D, E*) where Abf1 and Reb1 proteins directly contact DNA (*Jaiswal et al., 2016*; *McBroom and Sadowski, 1994a*). In contrast, 7meG damage levels were not affected outside of the core motif (e.g. –20 to –10 and 10–20 bp relative to the motif midpoint). 7meG levels were relatively even across the 'low-occupancy' Reb1 binding sites (*Figure 1F*), even though these sites have nearly identical motif sequence as the 'high-occupancy' binding sites. While damage formation was mainly suppressed in the core motif, we also saw increased 7meG levels (~1.5 fold) at a few positions (e.g. –7,–3, –2, and 0) at the edge of the Abf1 motif or between the two highly conserved regions within the motif (*Figure 1D* and *Figure 1—figure supplement 2*).

Repeat of the NMP-seq experiment generated reproducible damage mapping data (*Figure 1—figure supplement 3*) and showed inhibition of 7meG formation at Abf1 and Reb1 binding sites (*Figure 1—figure supplement 4*). Additionally, the reduction of damage formation was not due to deletion of the *MAG1* gene, because 7meG formation was also inhibited at Abf1 and Reb1 sites in WT-0 h cells after normalization to damage in the naked DNA (*Figure 1—figure supplement 5*).

Moreover, analysis of A reads indicates that 3meA formation was increased at the –3 position of the 'high-occupancy' Reb1 sites, but not at the same position of the 'low-occupancy' Reb1 sites (*Figure 1—figure supplement 6*). Intriguingly, the increased 3meA formation appears to be position dependent, because the adjacent –2 and –1 positions (both are conserved in A or T) did not show elevated 3meA damage formation. Analysis of the published Reb1-DNA complex structure (*Jaiswal et al., 2016*) indicates that Reb1 protein binding causes a large curvature (~56°) in DNA and significantly compresses the minor groove near the –3 position. These structural changes caused by Reb1 protein binding may play a role in modulating 3meA formation.

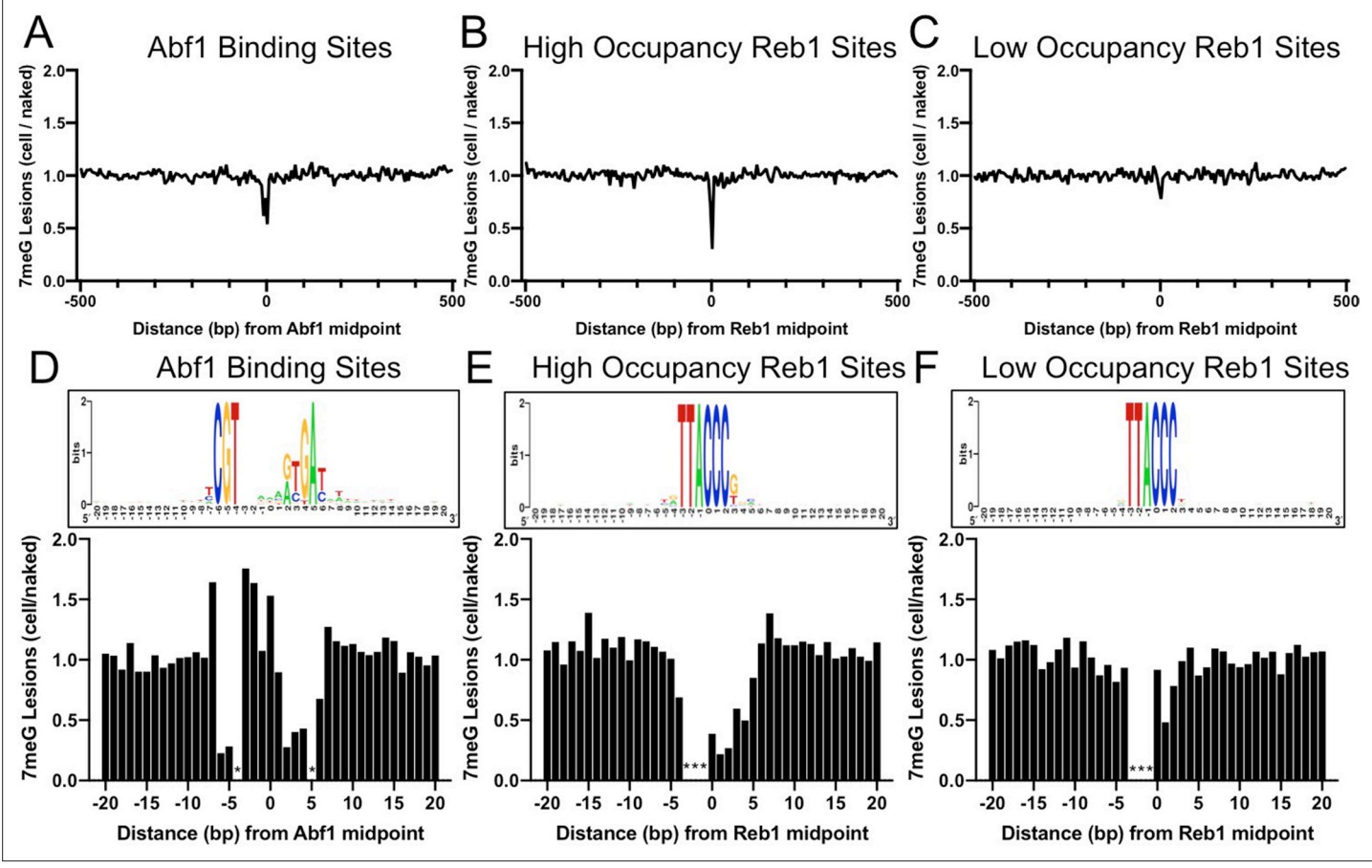

**Figure 1.** Formation of 7-methylguanine (7meG) lesions at ARS binding factor 1 (Abf1) and rDNA enhancer binding protein 1 (Reb1) binding sites. (**A**) Distribution of 7meG damage at 661 Abf1 binding sites and the flanking DNA in methyl methanesulfonate (MMS)-treated yeast cells. The cellular (i.e. *mag1Δ*–0 hr) 7meG levels in 5 bp non-overlapping moving windows were normalized to damage in naked yeast DNA. The normalized ratio was scaled to 1.0 and plotted along the aligned Abf1 sites. (**B**) Distribution of 7meG at 784 'high-occupancy' Reb1 binding sites and the flanking DNA. *N*-methylpurine-sequencing (NMP)-seq data was analyzed at Reb1 binding sites. (**C**) Distribution of 7meG at 472 'low-occupancy' Reb1 binding sites. (**D–F**) High-resolution plots showing 7meG formation in the Abf1, 'high-occupancy', and 'low-occupancy' Reb1 binding motif and the immediately adjacent DNA. The top panel depicts the consensus motif sequence for each transcription factor. The lower panel shows the normalized damage levels and each column points to a specific position at the binding site. Asterisks indicate conserved motif positions with exclusive A or T nucleotides and are not 7meG-forming sequences.

The online version of this article includes the following source data and figure supplement(s) for figure 1:

**Source data 1.** Source data for 7-methylguanine formation at Abf1 binding sites.

**Source data 2.** Source data for 7-methylguanine formation at Reb1 binding sites.

**Figure supplement 1.** NMP-seq methodology and damage mapping in naked yeast genomic DNA.

**Figure supplement 2.** Statistical analysis of 7meG formation at TF binding sites and the flanking DNA.

**Figure supplement 3.** Reproducibility of two independent NMP-seq datasets generated in the *mag1Δ* strain.

**Figure supplement 4.** Independent repeat of 7meG formation at Abf1 and Reb1 binding sites in *mag1Δ* cells.

**Figure supplement 5.** Formation of 7meG in wild-type (WT) cells.

**Figure supplement 6.** A hotspot of 3meA damage in the Reb1 motif.

## Abf1 and Reb1 binding inhibits repair of 7meG lesions

To address how TF binding affects 7meG repair in cells, we analyzed NMP-seq data generated after repair incubation (e.g. 1 and 2 hr repair). Repair analysis was conducted by normalizing 7meG lesions at each time point to the initial 7meG damage (i.e. 0 hr repair). This analysis considers the variable amounts of initial damage along the motif sequence, which can conceivably impact remaining damage

after repair. The normalization (i.e. damage after repair/initial damage) results in fraction of remaining damage, which is inversely correlated with DNA repair activity (*Mao et al., 2017*; *Mao et al., 2016*).

Our analysis indicates that repair of 7meG lesions was strongly suppressed at both Abf1 and Reb1 binding sites in wild-type (WT) cells, shown by peaks of unrepaired damage at 1 and 2hr (*Figure 2*) near the TF binding midpoint. The repair suppression is mediated by TF binding, not the underlying DNA sequence, because no repair inhibition was observed at 'low-occupancy' Reb1 binding sites (*Figure 2C, F*). Additionally, nucleosomes around the TF binding sites play an important role in affecting 7meG repair. Fast repair was observed in the nucleosome-depleted region around the TF binding site and linker DNA between two adjacent nucleosomes (*Figure 2*). In contrast, slow repair was found near nucleosome peaks, which is consistent with previous studies showing inhibition of BER at the nucleosome dyad center (*Kennedy et al., 2019*; *Mao et al., 2017*). These data indicate that BER is affected by the combined effects of TF binding and nucleosome positioning. Analysis of the remaining damage at single-nucleotide resolution confirmed repair inhibition and further indicated that repair was suppressed in an ~ 20–30bp DNA region for Abf1 and 'high-occupancy' Reb1, but not for 'low-occupancy' Reb1 binding sites (*Figure 2G-L*). Hence, TF binding inhibits BER in a slightly broader DNA region (both core motif and adjacent DNA) relative to its impact on NMP damage formation (mainly in the core motif).

Repair of 7meG by BER is initiated by the Mag1 glycosylase in yeast (*Wyatt et al., 1999*). To test if the inhibited repair of 7meG at TF binding sites is due to reduced BER, we analyzed 7meG repair in the *mag1Δ* mutant strain. NMP-seq analysis in this mutant revealed higher levels of unrepaired 7meG lesions at 2 hr than in WT (*Figure 2—figure supplement 1*), consistent with deficient BER for NMPs in the mutant. Moreover, there was no difference in remaining damage between the TF binding sites and flanking DNA in *mag1Δ* cells (*Figure 2—figure supplement 1*), confirming that BER is inhibited by TF binding.

## Depletion of Abf1 or Reb1 protein restores 7meG formation and elevates BER at their binding sites

Our data suggest that TF binding acts as a barrier to the damaging chemical MMS and BER enzymes. We hypothesize that removal of the TF would expose the binding sites to MMS and repair enzymes. As both Abf1 and Reb1 are essential for yeast survival and cannot be knocked out, we used the published Anchor-Away strategy (*Haruki et al., 2008*) to conditionally and rapidly export the protein from the nucleus to the cytoplasm. We then performed NMP-seq experiments in the TF-depleted yeast strains to analyze 7meG formation and repair. Both Abf1 and Reb1 anchor-away strains (Abf1-AA and Reb1-AA) were generated and used to study their impacts on gene transcription (*Kubik et al., 2018*; *Kubik et al., 2015*). We followed the published protocol to deplete Abf1 or Reb1 from the nucleus with rapamycin. Moreover, growth of Abf1-AA or Reb1-AA strain was inhibited on rapamycin-containing plates (*Figure 3—figure supplement 1*), confirming that nuclear depletion of either protein is lethal for yeast cells (*Kubik et al., 2015*).

In the control strain (WT-AA), in which no target protein is tagged for depletion, analysis of the NMP-seq data indicates that 7meG damage formation was still suppressed at the conserved motif sequences upon rapamycin treatment (*Figure 3—figure supplement 1*), indicating that rapamycin itself had little effect on NMP damage formation. However, TF depletion in Abf1-AA or Reb1-AA cells restored damage formation at their corresponding binding sites (*Figure 3—figure supplement 1*). For example, Abf1 depletion increased 7meG formation at Abf1 binding sites to a level comparable to the flanking DNA; however, no damage restoration was seen at Reb1 binding sites in Abf1-AA cells (*Figure 3—figure supplement 1*). Similarly, damage was restored at Reb1 binding sites in Reb1-AA cells, but not at Abf1 sites (*Figure 3—figure supplement 1*). Therefore, these data indicate that nuclear depletion of each TF specifically affects damage formation at its own binding sites, but has no effect on the binding sites of the other TF.

Analysis of 7meG repair in the AA strains indicates that BER was restored and even elevated by removing each TF from the binding sites. Compared to the control WT-AA strain (*Figure 3A, B*), no repair inhibition was seen at Abf1 binding sites when Abf1 was depleted (*Figure 3C*). Instead, BER was faster at Abf1 binding sites relative to the flanking DNA in Abf1-AA cells (*Figure 3C*), likely because these binding sites are located in nucleosome-depleted regions and damage is efficiently repaired by BER (*Mao et al., 2017*). Repair in the surrounding nucleosomes was also affected by Abf1

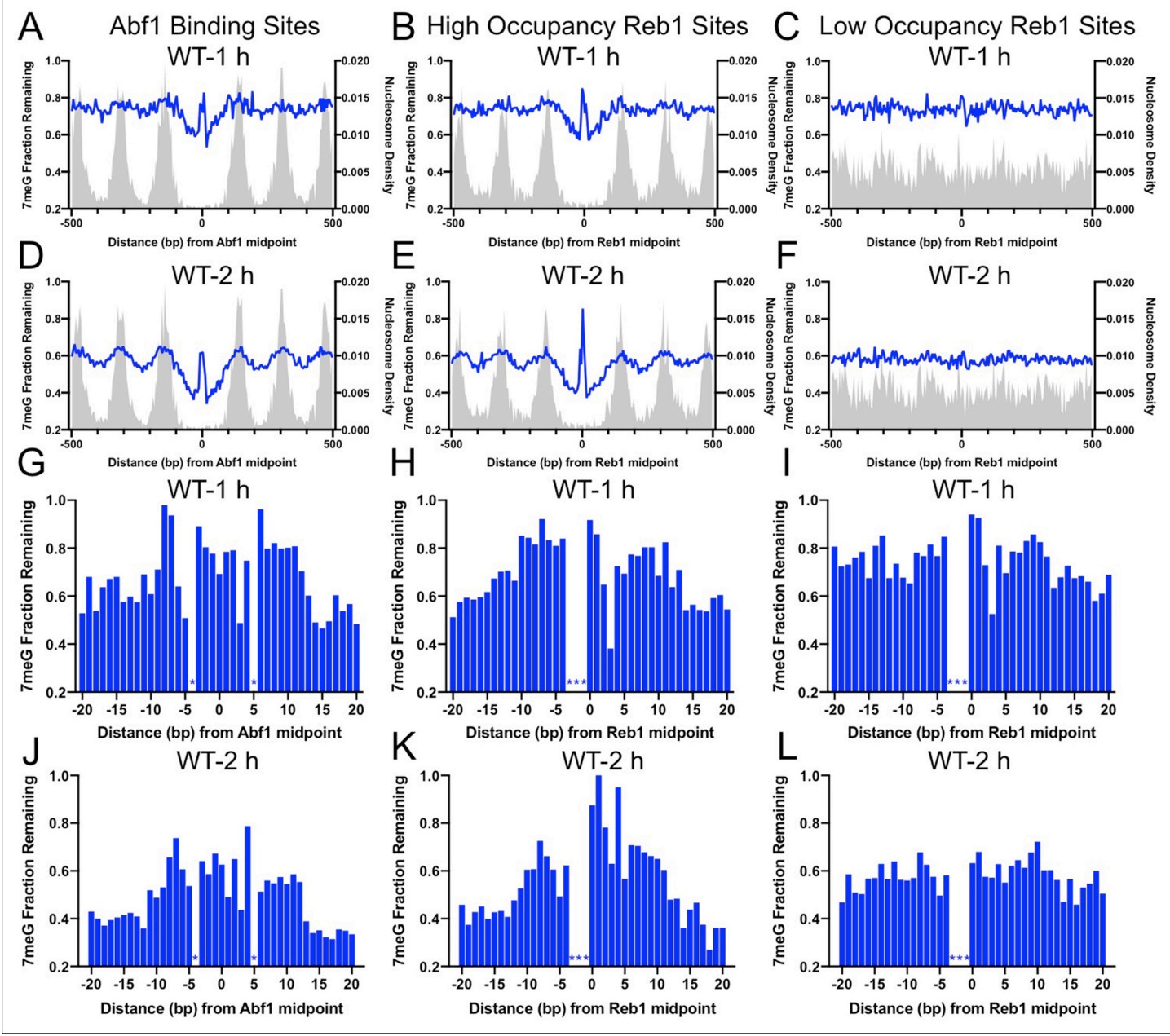

**Figure 2.** Base excision repair (BER) of 7meG lesions at ARS binding factor 1 (Abf1) and rDNA enhancer binding protein 1 (Reb1) binding sites. (**A**) The fraction of remaining 7meG lesions (blue line) after 1 hr repair at Abf1 binding sites in WT cells. Remaining 7meG at the binding sites and in the flanking DNA (up to 500 bp in each direction) was shown. The binding sites were obtained from the published ORGANIC method (*Kasinathan et al., 2014*). The plot shows the average remaining damage in 5 bp non-overlapping moving windows. The nucleosome density, which was analyzed using the published yeast MNase-seq data (*Weiner et al., 2015*), was plotted as the gray background. (**B**) Repair of 7meG lesions at 'high-occupancy' Reb1 binding sites. Data shows fraction of remaining damage at 1 hr. (**C**) Fraction of remaining 7meG at 'low-occupancy' Reb1 binding sites at 1 hr. (**D–F**) Fraction of remaining 7meG lesions after 2 hr repair for Abf1, 'high-occupancy' Reb1, and 'low-occupancy' Reb1 binding sites, respectively. (**G–I**) High-resolution analysis of remaining 7meG after 1 hr repair at Abf1, 'high-occupancy' Reb1, and 'low-occupancy' Reb1 sites, respectively. Damage remaining between –20 and 20 bp relative to the TF motif midpoint was shown. (**J**) to (**L**) high-resolution analysis of remaining 7meG after 2 hr repair.

The online version of this article includes the following source data and figure supplement(s) for figure 2:

**Source data 1.** Source data for remaining 7-methylguanine at Abf1 binding sites after 2 hr repair.

**Source data 2.** Source data for remaining 7-methylguanine at Reb1 binding sites after 2 hr repair.

**Figure supplement 1.** Repair of 7meG damage in base excision repair (BER)-deficient yeast cells (i.e. *mag1Δ*).

**Figure supplement 2.** Summary of DNA damage, base excision repair (BER), and their correlation with the Reb1–DNA complex structure.

**Figure supplement 3.** Inhibition of 7meG repair at transcription factor (TF) binding sites mapped by ChIP-exo.

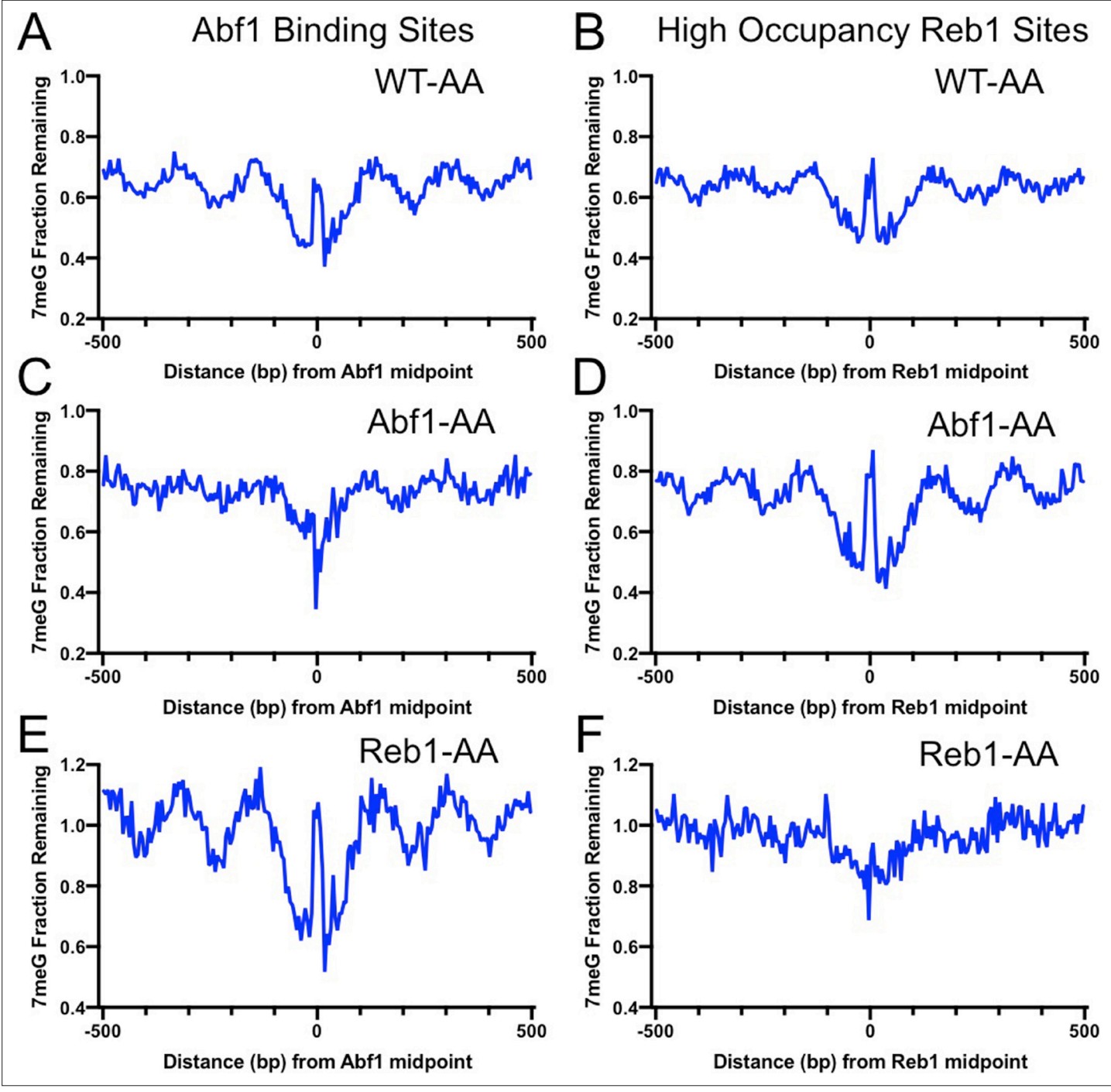

**Figure 3.** Base excision repair (BER) of 7meG lesions in anchor-away (AA) yeast strains. (**A**) Fraction of remaining 7meG lesions after 2 hr repair (normalized to 0 hr) in rapamycin-treated WT-AA cells at Abf1 binding sites. (**B**) Remaining 7meG at 'high-occupancy' Reb1 binding sites in rapamycin-treated WT-AA cells. (**C**) and (**D**) Fraction of remaining 7meG at 2 hr in Abf1-AA cells after rapamycin treatment at Abf1 and 'high-occupancy' Reb1 sites, respectively. (**E**) and (**F**) Remaining 7meG at 2 h in Reb1-AA cells after rapamycin-mediated protein depletion at Abf1 and Reb1 sites.

The online version of this article includes the following figure supplement(s) for figure 3:

**Figure supplement 1.** Depletion of Abf1 or Reb1 restores 7meG damage formation at the corresponding binding sites in yeast.

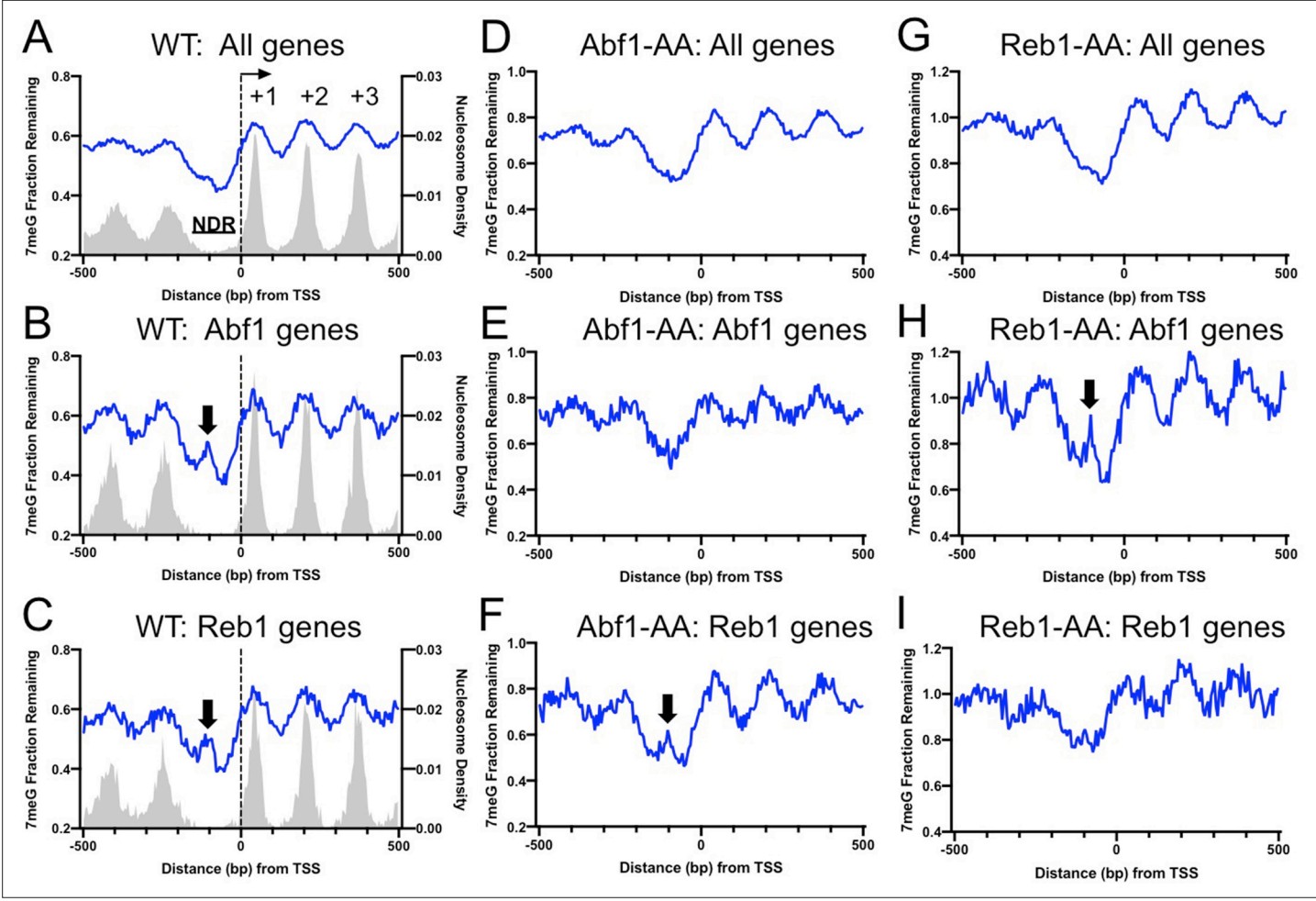

**Figure 4.** Repair of 7-methylguanine (7meG) in the ARS binding factor 1 (Abf1) and rDNA enhancer binding protein 1 (Reb1) target genes. (**A**) Average fraction of remaining 7meG lesions (blue line) after 2 hr repair in all yeast genes in WT cells. Genes (n = 5205) were aligned at the TSS (position 0) and repair was plotted in accordance with gene transcriptional direction. The average damage in 5 bp moving windows is shown from upstream 500 bp to downstream 500 bp relative to the TSS. The gray background indicates nucleosome peak density. (**B**) Average fraction of remaining 7meG lesions after 2 hr repair in WT cells in Abf1-linked genes (n = 697). (**C**) Average fraction of remaining 7meG lesions after 2 hr repair in WT cells in Reb1-linked genes (n = 708). (**D–F**) Fraction of remaining 7meG at 2 hr in Abf1-depleted cells for all genes, Abf1-linked, and Reb1-linked genes. (**G–I**) Fraction of remaining 7meG at 2 h in Reb1-depleted cells for all genes, Abf1-linked, and Reb1-linked genes.

The online version of this article includes the following figure supplement(s) for figure 4:

**Figure supplement 1.** Damage peaks overlap with Abf1 or Reb1 binding sites in gene promoters.

depletion (compare *Figure 3A, C*), potentially due to the weakened nucleosome organization around Abf1 binding sites in Abf1-AA cells (*Kubik et al., 2018*), further supporting that BER is affected by TF binding and the accompanying chromatin organization. Repair of 7meG damage was still inhibited at Reb1 binding sites in Abf1-AA cells (*Figure 3D*), consistent with Reb1 protein binding to its target sites in Abf1-AA cells. Similar results were also observed in Reb1-AA cells, where BER was inhibited at Abf1 binding sites (*Figure 3E*) but accelerated at Reb1 binding sites relative to the flanking DNA (*Figure 3F*). Taken together, these data demonstrate that removal of Abf1 and Reb1 exposes their target sites to the damaging chemical and BER enzymes.

## Abf1 and Reb1 inhibit BER in promoters of target genes

Abf1 and Reb1 bind to the nucleosome-depleted region (NDR) of gene promoters to facilitate transcription (*Kubik et al., 2018*). We next sought to understand how the two TFs affect BER in the context of gene transcription. We first examined the global BER pattern by analyzing 7meG repair in WT cells for all yeast genes. Genes (n = 5205) were aligned by their transcription start site (TSS) (*Park*

*et al., 2014*) and repair was analyzed in accordance with the transcriptional direction. As shown in *Figure 4A*, BER (average of all genes) was generally faster in NDR relative to the coding region where DNA is organized into + 1, + 2, and so on nucleosomes (*Figure 4A*), a pattern consistent with our previous studies (*Mao et al., 2017*). Hence, the global BER pattern revealed by our analysis indicates that Abf1 and Reb1 do not inhibit repair in NDR when all genes were included.

As Abf1 or Reb1 do not affect BER globally, we hypothesized that they may specifically affect BER in target genes. To test this hypothesis, we linked Abf1 and Reb1 binding sites to the closest TSS of annotated genes (*Park et al., 2014*). This association identified 697 Abf1-linked and 708 Reb1-linked genes (see Methods for detail). We then aligned Abf1-linked and Reb1-linked genes at their TSS and plotted 7meG repair in accordance with the transcriptional direction. For each subset of genes (i.e. Abf1- or Reb1-linked genes), we found a prominent damage peak in NDR after 2 hr repair in WT cells (*Figure 4B, C*, black arrows). The damage peak was located ~ 100 bp upstream of the TSS and overlapped with Abf1 or Reb1 binding peak (*Figure 4—figure supplement 1*). The association between repair inhibition and TF binding was further investigated in individual genes. To this end, we sorted Abf1- or Reb1-linked genes based on the distance between the TF binding site and the TSS (e.g. genes with longer distance shown on the top) and generated gene plots of remaining damage. Our gene-by-gene analysis revealed a strong correlation between BER inhibition (i.e. high remaining damage) and TF binding (*Figure 4—figure supplement 1*), indicating that Abf1 and Reb1 inhibit BER in their target genes.

This finding was further confirmed by analyzing NMP-seq data generated in the AA cells. We found that depletion of Abf1 in Abf1-AA cells did not change the global BER pattern when all genes were included (*Figure 4D*), but it restored repair in the NDR of Abf1 target genes (*Figure 4E*). As expected, repair in Reb1 target genes was still inhibited in the Abf1-AA cells (*Figure 4F*, black arrow). Similar results were found in the Reb1-AA cells (*Figure 4G-I*). The damage peaks in NDR were not as high as repair analysis at the mapped TF binding sites (e.g. compare *Figure 4B* with *Figure 2D*), likely because the gene analysis was performed in each subset of genes aligned on their TSS, not the midpoint of the TF binding sites.

## Repair of 3meA is inhibited by TF binding in vivo and in vitro

Although 3meA is much less abundant than 7meG in MMS-treated cells, 3meA has long been known to be cytotoxic (*Fu et al., 2012*; *Plosky et al., 2008*). Conventional methods studying cellular repair of MMS-induced damage (e.g. AAG/APE1 digestion followed by gel electrophoresis) (*Czaja et al., 2014*) cannot distinguish repair of 7meG and 3meA. Additionally, 3meA is unstable and difficult to be synthesized in vitro. As NMP-seq maps both 3meA and 7meG lesions, we extracted A reads to specifically analyze 3meA repair.

Analysis of 3meA lesions in WT cells indicates that the repair was inhibited near the center of Abf1 and 'high-occupancy' Reb1 binding sites, as shown by high levels of remaining 3meA lesions at 2 hr (*Figure 5A, B*). In contrast, 3meA repair was not inhibited at 'low-occupancy' Reb1 binding sites (*Figure 5C*). Interestingly, the 3meA peaks appear to be narrower than the 7meG peaks, and no clear 3meA repair inhibition was seen in nucleosomes surrounding the TF binding sites. These differences are consistent with the greater activity of Mag1 and its homologs in removing 3meA than 7meG (*Connor et al., 2005*), which may lead to less repair inhibition to 3meA lesions by DNA-binding proteins.

A closer examination of 3meA repair at 'high-occupancy' Reb1 binding sites revealed a slow repair spot at the +4 position (*Figure 5—figure supplement 1*). Repair of 7meG was also inhibited at the same location (*Figure 5—figure supplement 1*), suggesting that the +4 position is refractory to BER enzymes. Although the sequence at +4 position is not conserved in the Reb1 motif, the Reb1–DNA crystal structure (*Jaiswal et al., 2016*) shows that this position is directly contacted by the DNA binding domain of Reb1 protein.

The strong repair inhibition at the + 4 position led us to further investigate BER using an in vitro system. To simulate 3meA repair at the Reb1 binding site, we incorporated a stable AAG substrate, inosine (denoted as I), at the + 4 position of the motif strand (*Figure 5D*). Inosine can naturally arise from adenine deamination in cells and is repaired by AAG-mediated BER (*Alseth et al., 2014*). We found that inosine incorporation did not significantly change Reb1 binding affinity compared to DNA without inosine (*Figure 5—figure supplement 1*). AAG and APE1 enzymes were added to naked

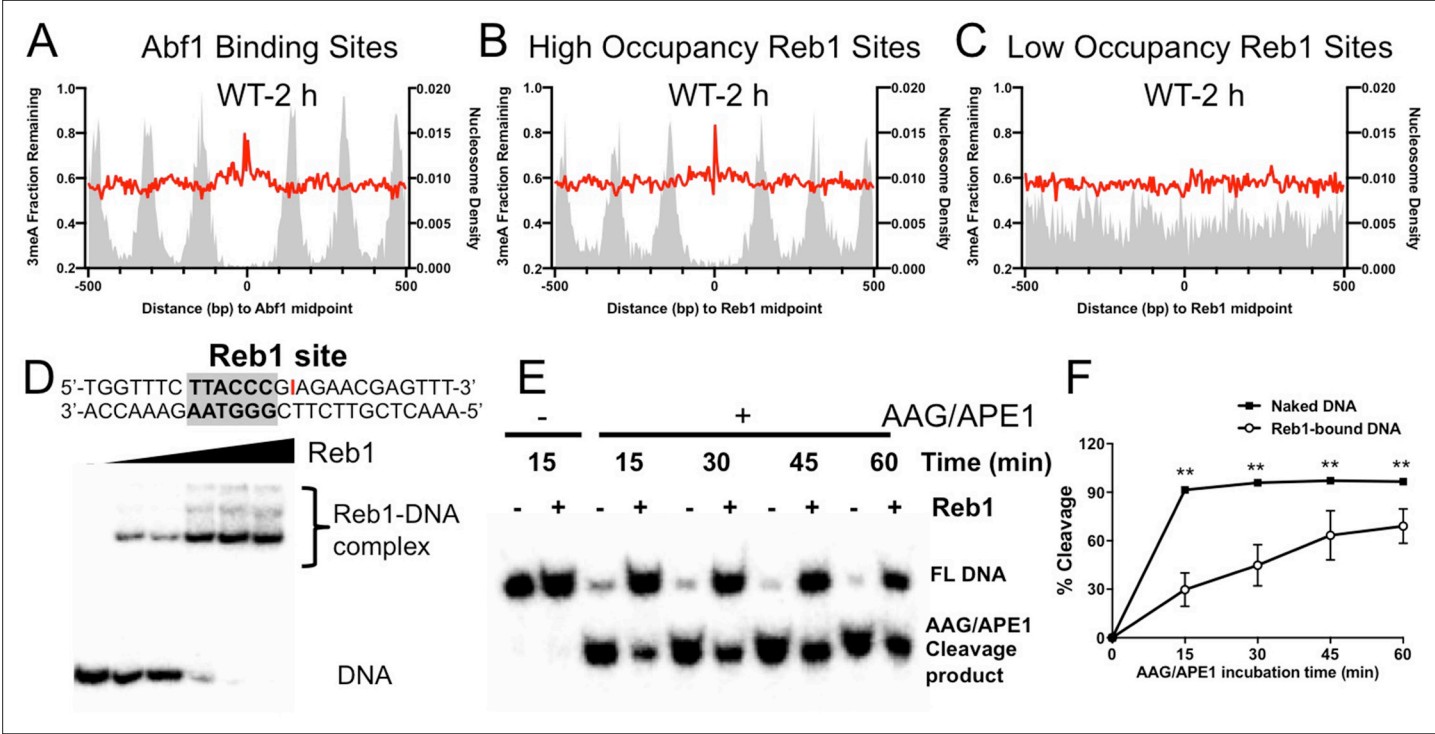

**Figure 5.** Repair of 3-methyladenine (3meA) at transcription factor (TF) binding sites. (**A**) Average fraction of remaining 3meA lesions (red line) at Abf1 binding sites mapped with the ORGANIC method. Data shows fraction of remaining 3meA lesions in 5 bp non-overlapping moving windows along the binding sites in WT cells at 2 hr. (**B**) and (**C**) fraction of remaining 3meA damage at 'high-occupancy' and 'low-occupancy' Reb1 binding sites, respectively. (**D**) The upper panel shows synthesized double-stranded DNA containing a Reb1 binding site. The inosine damage (red) was incorporated at the +4 position on the Reb1 motif strand. The lower panel shows gel shift data with DNA alone or DNA incubated with increasing amounts of purified Reb1 protein. DNA was labeled with $^{32}$P on the 5' end of the motif strand. (**E**) Cleavage of the inosine-containing DNA or DNA complexed with Reb1 protein by AAG/APE1 enzymes. The substrates (naked DNA or DNA-Reb1 complex) were incubated with AAG and APE1 enzymes to cleave the damage site. DNA was analyzed on denaturing polyacrylamide gels to separate the full-length DNA (FL DNA) and the cleavage product. (**F**) Quantification of the repair gels. Graph shows the percent of cleaved DNA (lower band) relative to total DNA (lower and upper bands) at different incubation time points. Average cleavage and standard deviation from four independent repair experiments are shown (**p<0.005 by t-tests).

The online version of this article includes the following source data and figure supplement(s) for figure 5:

**Source data 1.** Unedited gel shift data showing binding of Reb1 protein to DNA.

**Source data 2.** Cleavage of the inosine-containing DNA or DNA complexed with Reb1 protein.

**Source data 3.** Gel shift data showing Reb1 binding to undamaged (top) and damaged (bottom) DNA.

**Source data 4.** Gel shift data showing Reb1 binding to DNA (damage is incorporated into the bottom strand).

**Source data 5.** Cleave of inosine-containing DNA (naked DNA) or DNA bound by Reb1 by AAG/APE1.

**Figure supplement 1.** Repair inhibition at the +4 position of Reb1 binding sites.

DNA or DNA pre-bound with purified Reb1 protein to examine BER activity in vitro. The AAG/APE1 cleavage product (i.e. the lower band) was analyzed in a time-course experiment to compare BER activity between free DNA and Reb1-bound DNA (*Figure 5E*). Quantification of the gels showed significantly reduced repair activity at the binding site in Reb1-bound DNA relative to the naked DNA substrate (*Figure 5F*). Reduced BER activity was also observed when inosine was placed on the other strand at the + 4 position (*Figure 5—figure supplement 1*). Hence, these in vitro data, consistent with our cellular damage sequencing data, indicate that BER of DNA base damage is suppressed by TF binding.

## TF binding inhibits both BER and NER

TF binding has been shown to inhibit NER of UV damage (*Frigola et al., 2021*; *Hu et al., 2017*); however, it is not known if NER and BER are inhibited to the same extent. Using a UV damage mapping method cyclobutane pyrimidine dimer sequencing (CPD-seq), we previously showed that formation

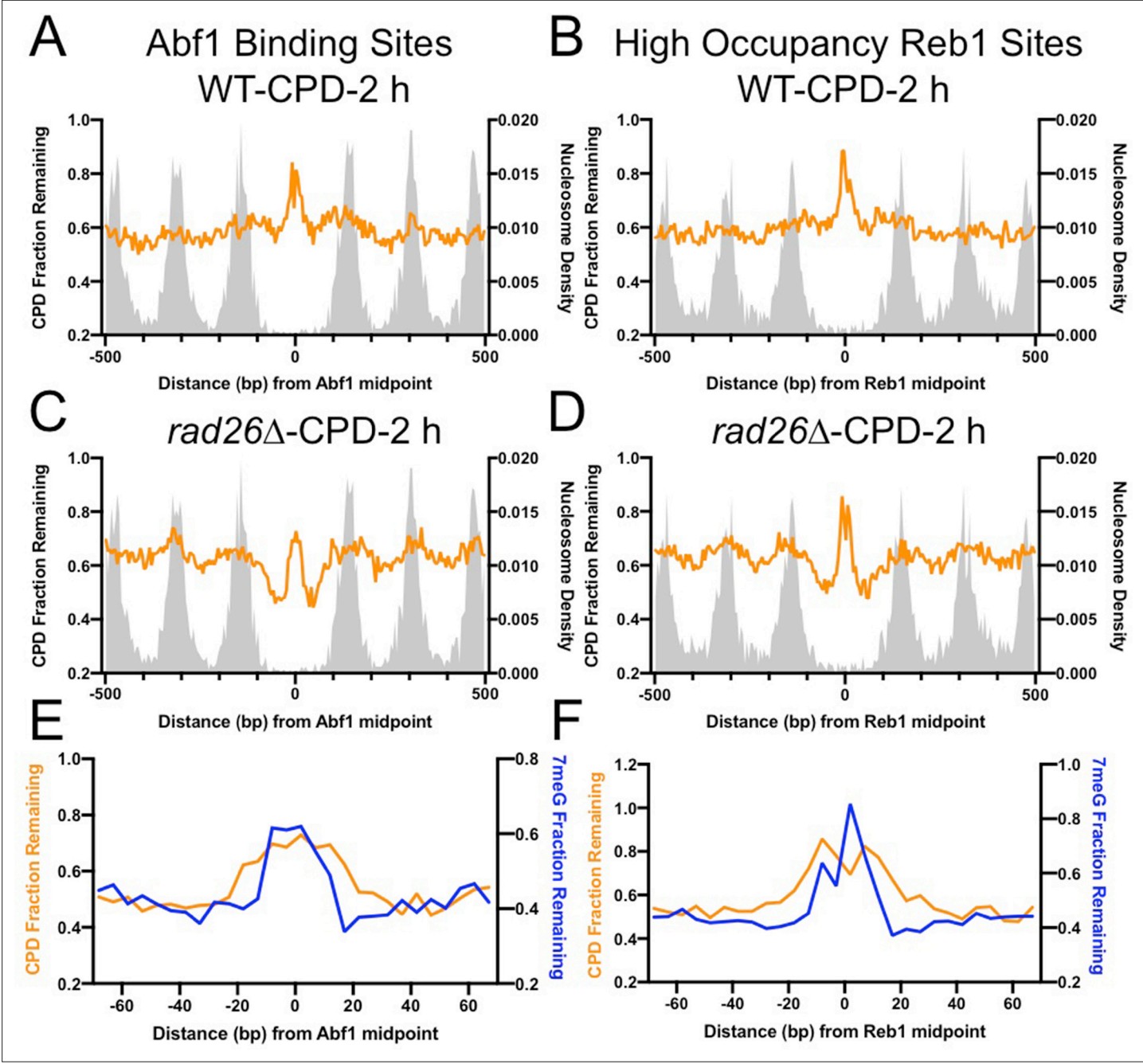

**Figure 6.** Comparison of cyclobutane pyrimidine dimer (CPD) and 7-methyladenine (7meG) repair at transcription factor (TF) binding sites. (**A**) Fraction of remaining CPDs at Abf1 binding sites in WT-2 hr cells. Similar to NMP-seq data analysis, the number of CPD-seq reads at 2 hr was normalized to initial damage reads at 0 hr. The resulting fraction of remaining CPDs was plotted at Abf1 binding sites and flanking DNA up to 500 bp. The average remaining damage in 5 bp non-overlapping moving windows was shown. (**B**) Fraction of remaining CPDs was analyzed at 'high-occupancy' Reb1 binding sites. (**C**) Fraction of remaining CPDs at Abf1 binding sites in the *rad26Δ* mutant strain, in which CPD repair is mainly conducted by GG-NER. (**D**) Fraction of remaining CPDs at Reb1 binding sites in the *rad26Δ* mutant cells. (**E**) and (**F**) comparison between GG-NER (orange line) and BER (blue line) at Abf1 and Reb1 binding sites, respectively. GG-NER was analyzed using CPD-seq data (2 hr relative to 0 hr) generated in *rad26Δ* cells. BER analysis was conducted with NMP-seq data.

of UV-induced CPDs is significantly suppressed at Abf1 and Reb1 binding sites (*Mao et al., 2016*). To investigate NER of CPDs at Abf1 and Reb1 binding sites, we analyzed CPD-seq data generated in UV-irradiated yeast cells. We found that repair of CPDs at 2 hr (normalized to CPDs at 0 hr) was inhibited at both Abf1 and Reb1 binding sites in WT cells, shown by high levels of unrepaired CPDs at

the binding sites relative to the flanking nucleosome-occupied DNA (*Figure 6A, B*). As both Abf1 and Reb1 binding sites are localized in gene promoters (*Figure 4—figure supplement 1*), transcription-coupled NER (TC-NER) may play a role in the removal of CPDs in transcribed regions surrounding the binding sites. To reduce the interference from TC-NER, we analyzed CPD-seq data generated in a *rad26Δ* mutant strain in which TC-NER is severely diminished (*Duan et al., 2020*), thus allowing us to focus on global genomic NER (GG-NER). Our data indicates that GG-NER was suppressed at the center of Abf1 and Reb1 binding sites, but elevated in DNA adjacent to the center due to depletion of nucleosomes (*Figure 6C, D*), similar to the BER pattern (*Figure 2A, B*). Additionally, GG-NER was also modulated by nucleosomes positioned around the TF binding sites. These analyses indicate that GG-NER is inhibited by both Abf1 and Reb1 at their binding sites.

As the GG-NER pattern at the TF binding sites resembles the BER pattern revealed by our NMP-seq data, we sought to understand if the size of the inhibited DNA region is the same for both repair pathways. A comparison between CPD and 7meG repair indicates that GG-NER was inhibited in a broader DNA region at Abf1 and Reb1 binding (*Figure 6E, F*). While BER (i.e. 7meG repair) was inhibited in ~ 30 bp DNA surrounding the center of the binding motif, inhibition of GG-NER was extended by an additional 10 bp on each side (*Figure 6E, F*). These high-resolution sequencing data demonstrate the difference between BER and GG-NER at TF binding sites, which is consistent with the different mechanisms underling NER and BER (see Discussion).

## Discussion

In this study, we used MMS-induced damage as a model lesion and analyzed base damage distribution and BER at the binding sites of yeast TFs. Our high-resolution damage mapping data revealed an important role for TF binding in modulating initial damage formation and inhibiting BER. As base damage (e.g. oxidative, alkylation, uracil, and AP sites) has long been recognized as an important source of DNA mutations in human cancers (*Maynard et al., 2009*; *Tubbs and Nussenzweig, 2017*), the interplay between TF binding, base damage formation, and BER revealed by our study has important implications for understanding mutations in gene regulatory regions.

Our data shows that TF binding can significantly modulate NMP damage formation. Depending on the location and the conservation level in the binding motif, TF binding can both suppress and elevate damage levels. The highly conserved nucleotides in the core motif of both Abf1 and Reb1 binding sites mainly suppressed NMP damage formation (*Figure 1*). NMP damage is formed via the chemical reaction between the alkylating agent and individual nucleotides (*Fu et al., 2012*). Due to protein–DNA interactions, nucleotides with restrained reactivity with MMS will be less sensitive and thus generate reduced amounts of damage. The highly conserved nucleotides in the Abf1 and Reb1 motifs are directly contacted by specific amino acids of the protein (*Jaiswal et al., 2016*; *McBroom and Sadowski, 1994a*). This suggests that Abf1 and Reb1 reduce the reactivity of the bound nucleotides, thus protecting conserved parts of the core motif from alkylation DNA damage. The protective role of TFs does not seem be specific for alkylation damage. UV damage formation was also reported to be suppressed by TF binding in yeast (*Mao et al., 2016*) and human cells (*Frigola et al., 2021*). Thus, TF binding may function as an important mechanism in cells that protects conserved regulatory sequences from being damaged and mutated.

A few specific positions in the Abf1 motif exhibit elevated 7meG formation. Moreover, we also found a specific 3meA hotspot in the Reb1 motif. While the detailed mechanism for elevated alkylation damage formation is unclear, previous studies of UV damage formation revealed that TF binding-mediated DNA structural change plays a critical role in dictating damage yields. Indeed, human ETS (E26 transformation-specific) TFs have been shown to change the DNA geometry at their binding sites and cause individual UV damage and mutation hotspots (*Elliott et al., 2018*; *Mao et al., 2018*). Yeast Abf1 protein has been shown to bend DNA toward its minor groove (*McBroom and Sadowski, 1994b*). DNA bending caused by Abf1 may expose certain bases in the motif and increase their reactivity with MMS, resulting in elevated damage yields.

The published complex structure of Reb1 (from *Schizosaccharomyces pombe*) with DNA (*Jaiswal et al., 2016*) provided an opportunity to investigate how TF–DNA interactions could modulate alkylation damage distribution and BER (*Figure 2—figure supplement 2*). The DNA-binding domain (DBD) of Reb1 winds around two turns of duplex DNA as a series of four helix-turn-helix (HTH) domains, forming a so-called "saddle"-shaped structure (*Figure 2—figure supplement 2*). Two homologous

HTH domains, termed MybAD1 and MybAD2, are followed by two homologous repeat domains MybR1 and MybR2. C-terminal to the DBD is a transcription termination domain (TTD) that is not essential to DNA binding (*Jaiswal et al., 2016*). Within the central core of the Reb1 consensus (5'-GGGTAA-3'; the underlined G is position 0), positions +2–0 (i.e., GGG) are directly bound by both MybAD2 and MybR1 and exhibit significantly reduced 7meG formation (*Figure 2—figure supplement 2*). Positions −1 to −3 (i.e. TAA) are sandwiched between the subsites for MybAD1 and MybAD2, which insert recognition helices into the adjacent DNA major groove. As a result, the minor groove from positions −1 to −3 is strongly compressed in width and increased in depth (*Figure 2—figure supplement 2*). These results suggest that preferential formation of 3meA at position −3 may be facilitated by enhanced minor groove narrowing and DNA curvature by Reb1 binding. As expected, the strong binding of DBD leads to significantly slower BER kinetics in the central core, as shown by little change of 7meG damage levels after 1 or 2 hr repair compared to the initial damage at 0 hr (*Figure 2—figure supplement 2*). In addition to the core motif, the structural data indicates that nucleotides in the flanking DNA are bound by the Reb1 protein (*Jaiswal et al., 2016*). Consistently, BER in the flanking regions (e.g. −12 to −4 and +4 to +12) was also inhibited (*Figure 2—figure supplement 2*). Although Abf1–DNA complex structure data is currently unavailable, it is conceivable that Abf1 also binds to both the core motif and part of the flanking DNA. The strength of protein–DNA interaction in the flanking DNA may not be as high as in the core motif, which still allows damage formation to occur, but it considerably reduces the access of BER enzymes, particularly in DNA immediately adjacent to the core motif. As BER is generally inhibited in TF-bound DNA, damage hotspots induced by TF binding cannot be efficiently repaired and may eventually cause individual mutation hotspots when DNA is replicated. Considering the conserved damage formation and repair mechanisms between yeast and human cells, our findings provide a potential explanation to mutation hotspots at TF binding sites in non-UV exposed tumors. Moreover, recent studies suggest that AAG-mediated BER pathway plays a direct role in affecting RNA Pol II transcription (*Montaldo et al., 2019*). This, in combination with mutations at TF binding sites, may affect gene expression in cells exposed to alkylating agents.

The comparison between NMP and CPD repair at TF binding sites provides new insights into how TFs affect BER and NER differently. While TF binding inhibits both BER and NER, we found that the affected DNA region is considerably broader in NER compared to BER. NER is inhibited in about 50 bp DNA centered on the midpoint of Abf1 or Reb1 binding sites, whereas BER is suppressed in a narrower DNA region (*Figure 6*). The extended inhibition region in NER is consistent with more proteins being involved in NER compared to BER and more stable multi-protein complex assembled on DNA for NER. Moreover, NER requires repair endonucleases to cleave upstream of the 5' side and downstream of the 3' side relative to the lesion, releasing a repair intermediate of ~ 25 nt (*Huang et al., 1992*; *Schärer, 2013*). Although UV damage located outside of the TF binding site may be recognizable by the damage recognition factor such as XPC or yeast Rad4, one of the two repair cleavage sites may still be located within the binding motif and is inaccessible to the repair endonuclease. Hence, the unique 'dual-incision' mechanism of NER is consistent with the broader repair-resistant DNA region around a TF binding site compared to BER.

In summary, we generated high-resolution alkylation damage and BER maps at yeast TF binding sites, which allows us to elucidate how TF binding modulates base damage formation and repair. Considering the potential connection between base damage, BER, and mutations in non-UV exposed tumors, these analyses provide important insights into cancer mutations frequently elevated at TF binding sites.

## Materials and methods
### Yeast strains
WT and *mag1Δ* strains were in the BY4741 background. The AA strains, including WT-AA, Abf1-AA, and Reb1-AA, were gifts from Dr. David Shore (*Kubik et al., 2018*; *Kubik et al., 2015*).

### MMS treatment
Yeast cells were grown in YPD (yeast extract-peptone-dextrose) medium to mid-log phase and treated with 0.4% (v/v) MMS (Acros Organics, AC15689) for 10 min to induce alkylation damage. Cells were

centrifuged and washed with sterile deionized water to remove MMS. Cells were resuspended in pre-warmed YPD medium and incubated for repair in a 30°C shaker.

The AA yeast cells were pre-treated with 1 µg/ml rapamycin (Thermo Fisher Scientific, NC0678468) for 1 hr in YPD medium, as described in previous studies (*Haruki et al., 2008*; *Kubik et al., 2018*). At the end of rapamycin treatment, MMS was added to the culture and incubated for 10 min. After MMS treatment, cells were spun down and washed with sterile water to remove MMS. Cells were then resuspended in fresh YPD containing 1 µg/ml rapamycin for repair time points.

To damage naked yeast DNA with MMS, genomic DNA was first isolated from WT yeast cells without MMS treatment. All proteins were removed during DNA isolation by using vigorous phenol chloroform extraction, followed by ethanol precipitation. The purified DNA was incubated with MMS for 10 min. After MMS treatment, DNA was purified by phenol chloroform extraction and ethanol precipitation.

## NMP-seq library Preparation

NMP-seq library preparation was described in our previous study (*Mao et al., 2017*). Genomic DNA was sonicated to small fragments and ligated to the first adaptor DNA. The ligation product was purified and incubated with terminal transferase and dideoxy-ATP (ddATP) to block all free 3' ends (*Ding et al., 2015*). The NMP lesion site was cleaved by hAAG (NEB, M0313S) and APE1 (NEB, M0282S) to generate a new ligatable 3' end. DNA was denatured at 95°C and cooled on ice, followed by ligation to the second adaptor. After purification with Streptavidin beads (Thermo Fisher Scientific, 11,205D), the library DNA was briefly amplified by PCR with two primers complementary to the two adaptors. Sequencing of NMP-seq libraries was conducted on an Iron Torrent platform. Each NMP-seq experiment was performed twice independently and data reproducibility was tested with Pearson correlation coefficient analysis (*Figure 1—figure supplement 2*).

## TF binding data Sets

We used published yeast TF binding data sets in this study. These analyses were performed using the published ORGANIC binding data (*Kasinathan et al., 2014*). Binding sites were obtained from experiments using 10 min micrococcal nuclease digestion with 80 mM NaCl, as described in our previous study (*Mao et al., 2016*). Only binding sites with the canonical Abf1 or Reb1 motif sequence (CGTNNNNNRNKA and TTACCC, respectively) were used for damage and repair analysis. Binding sites that did not match the motif sequences were excluded. Reb1 binding sites were further stratified into 'high-occupancy' (occupancy > 10) and 'low-occupancy' (occupancy ≤ 10) binding sites based on the mapped occupancy levels (*Kasinathan et al., 2014*).

To identify target genes for Abf1 and Reb1, we searched gene TSS to find the closest midpoint of Abf1 or Reb1 binding sites using the ORGANIC datasets. If the TF binding site is located within 300 bp upstream or downstream of the gene TSS, the gene is identified as a putative target gene. Some binding sites are located in the middle of two divergently transcribed genes. In this case, both genes are recognized as target genes.

## NMP-seq data analysis

Analysis of NMP-seq datasets was conducted using our published protocols (*Mao et al., 2017*). NMP-seq sequencing reads were demultiplxed and aligned to the yeast reference genome (sacCer3) using Bowtie 2 (*Langmead and Salzberg, 2012*). For each mapped read, we identified the position of its 5' end in the genome using SAMtools (*Li et al., 2009*) and BEDTools (*Quinlan and Hall, 2010*). Based on the 5' end position, the single nucleotide immediately upstream of the 5' end was found and the sequence on the opposing strand was identified as the putative NMP lesion. The number of sequencing reads associated with each of the four nucleotides (e.g. A, T, C, and G) was counted to estimate the enrichment of MMS-induced NMP lesions in the sequencing libraries. G reads were typically highly enriched relative to C reads, followed by A reads.

To analyze damage formation and BER at TF binding sites, we extracted G or A reads to analyze 7meG and 3meA lesions, respectively. The number of lesions at each position around the midpoint of Abf1 or Reb1 binding sites was counted using the BEDTools intersect function. For damage formation, the cellular lesion counts were normalized to the naked DNA to account for the impact of DNA sequences on NMP lesion formation. The normalized ratio was scaled to 1.0 and plotted along the

TF binding sites (e.g. *Figure 1A-C*). Plots at single nucleotide resolution (e.g. *Figure 1D-F*) also show scaled damage ratio between cellular and naked DNA NMP-seq data. For repair analysis, damage counts at repair time points were normalized to the initial damage at 0 hr to generate fraction of remaining damage. Positions with high fraction of remaining damage are indicative of slow repair, since a large fraction of damage is not repaired at that site. Some highly conserved positions at TF binding sites do not have lesion-forming nucleotides. These positions are labeled with asterisks in single-nucleotide resolution plots (e.g. *Figure 1D-F*). Alternatively, we analyzed the average damage in a 5 bp non-overlapping moving window to show the average damage and repair in a broader DNA region (e.g. *Figures 1A and 2A*).

Some NMP-seq datasets such as mag1-0 hr, WT-1 hr, and WT-2 hr, were downloaded from our published studies (NCBI GEO, accession code GSE98031). New NMP-seq data generated in this study, including NMP data in naked DNA and in anchor-away yeast strains, have been submitted to NCBI GEO (accession code GSE183622). In some of the new samples (e.g. WT-AA, Abf1-AA-rep 2), we tried to add MMS-damaged pUC19 plasmid as spike-in control to quantify repair efficiency. Hence, the fraction of remaining damage in these samples was normalized by the pUC19 read ratio between 0 hr and 2 hr.

## CPD-seq datasets and analysis

Yeast CPD-seq data were downloaded from NCBI GEO (accession code GSE145911). Analysis of CPD repair at Abf1 and Reb1 binding sites was performed using the same method described in NMP-seq data analysis.

## In vitro Reb1 binding and BER assay

Recombinant Reb1 protein was expressed in *Escherichia coli* cells in a pET30a(+) expression vector (a gift from Dr. David Donze at Louisiana State University). Protein was purified with Co-NTA resin and eluted using 0.25 M imidazole. The purity of the eluted protein was ~90% as judged by Coomassie-stained SDS-PAGE. The nominal molecular weight of the recombinant construct was ~55 kDa. Protein concentration was determined by UV absorption at 280 nm. Reb1-DNA binding was analyzed using electrophoretic mobility shift assay (EMSA). Inosine lesion containing oligonucleotide (40 µM), or control oligonucleotide without inosine, was labeled with γ-$^{32}$P ATP (20 µCi) (Perkin Elmer) in a 25 µl reaction containing 1× PNK buffer and 15 units of polynucleotide kinase (New England Biolabs) by incubating at 37°C for 45 min. The reaction was heat inactivated at 65°C for 15 min. G-25 sephadex G-50 DNA grade resin columns were used to remove unincorporated γ-$^{32}$P ATP according to manufacturer's instructions (illustra GE Healthcare). The purified strand was used for subsequent annealing with equal amount of complementary strand in 50 µl total volume. The annealed duplex DNA (20 pmol) was mixed with increasing concentrations of Reb1 (0, 5.5, 11, 22, 33 and 44 pmol) in 50 µL reactions in 1× EMSA buffer containing 160 µg/ml BSA. The binding reaction was incubated on ice for 40 min. Free DNA and DNA bound by Reb1 were loaded onto an 8% native PAGE and separated by gel electrophoresis at 200 V for 30 min. The gel was exposed to a phosphor screen and the image was scanned using a Typhoon FLA7000 scanner (GE Healthcare). Gel quantification was performed with the ImageQuant software (GE Healthcare).

For BER assays, equal amount of naked DNA and DNA bound by Reb1 protein (~5 pmol of DNA) were incubated with AAG (~10 units) and APE1 (~1 unit) (New England Biolabs) in a 20 µl reaction at 37°C for 15, 30, 45, and 60 min. BSA (100 µg/ml), carried over from the EMSA step, was included in all BER reactions, including naked DNA without Reb1 protein. After BER cleavage, DNA was purified using Phenol:Chloroform:Isoamyl alcohol extraction and precipitated using ethanol. The purified DNA was resuspended in formamide (80%) and denatured at 95°C for 10 min. The denatured DNA was analyzed by electrophoresis at 200 V for 30 min using 12% polyacrylamide urea gels. The gel was exposed to a phosphor screen and imaged using a Typhoon FLA7000 scanner and quantified by ImageQuant.

## Structural analysis

The co-crystal structure of *S. pombe* Reb1 with terminator DNA that harbors a core consensus 5'-GGGTAA-3' (PDB: 5eyb) was used (*Jaiswal et al., 2016*). The bound DNA was analyzed using curves+ (*Lavery et al., 2009*) to fit the helical curvature and groove parameters. Values of helical

parameters were reported as averages ± standard deviations for the two copies found in the asymmetric unit. Atom-centered electrostatic potentials at 25°C in implicit water were computed using APBS (*Baker et al., 2001*) based on atomic charges and radii assigned from the AMBER14 forcefield. The solute dielectric was set to eight based on recently reported measurements on duplex DNA (*Cuervo et al., 2014*).

## Acknowledgements

We thank Mark Wildung and Wei Wei Du for technical assistance with Ion Proton sequencing. We also thank Dr. David Shore for providing anchor-away yeast strains. This work was supported by National Institute of Environmental Health Sciences Grants R21ES029302 (to P.M. and J.J.W.), R01ES032814 and R01ES028698 (to J.J.W.), NSF grant MCB 2028902 (to G M K P), and a pilot grant from UNM Center for Metals in Biology and Medicine (P20GM130422). This research was partially supported by UNM Comprehensive Cancer Center Support Grant NCI P30CA118100 and UNM ATG Shared Resource.

## Additional information

### Funding

| Funder | Grant reference number | Author |
| --- | --- | --- |
| National Institute of Environmental Health Sciences | R21ES029302 | John J Wyrick Peng Mao |
| National Institute of Environmental Health Sciences | R01ES032814 | John J Wyrick |
| National Institute of Environmental Health Sciences | R01ES028698 | John J Wyrick |
| National Science Foundation | MCB 2028902 | Gregory MK Poon |
| National Institute of General Medical Sciences | P20GM130422 | Peng Mao |
| National Cancer Institute | P30CA118100 | Mingrui Duan Peng Mao |

The funders had no role in study design, data collection and interpretation, or the decision to submit the work for publication.

### Author contributions

Mingrui Duan, Data curation, Methodology, Software, Writing – review and editing; Smitha Sivapragasam, Jacob S Antony, Jenna Ulibarri, Data curation, Writing – review and editing; John M Hinz, Supervision, Writing – review and editing; Gregory MK Poon, Formal analysis, Methodology, Writing – review and editing; John J Wyrick, Conceptualization, Data curation, Formal analysis, Methodology, Supervision, Validation, Writing – review and editing; Peng Mao, Conceptualization, Data curation, Formal analysis, Funding acquisition, Investigation, Methodology, Project administration, Resources, Software, Supervision, Validation, Writing - original draft, Writing – review and editing

### Author ORCIDs

Mingrui Duan ⓘ http://orcid.org/0000-0002-2352-1840
Smitha Sivapragasam ⓘ http://orcid.org/0000-0002-5599-9988
Jacob S Antony ⓘ http://orcid.org/0000-0002-1481-9768
Peng Mao ⓘ http://orcid.org/0000-0003-2068-1344

### Decision letter and Author response

Decision letter https://doi.org/10.7554/eLife.73943.sa1

Author response https://doi.org/10.7554/eLife.73943.sa2

## Additional files

### Supplementary files

• Transparent reporting form

• Source code 1. Mapping NMP-seq reads to yeast reference genome and identify the lesion site. The code consists of the bioinformatics pipeline using bowtie2, samtools, and bedtools to identify NMP lesions in the yeast genome.

• Source code 2. Intersect NMP-seq lesions with yeast nucleosomes and TFBSs. The code uses the intersect function of bedtools to identify NMP lesions in yeast nucleosomes and TF binding sites.

### Data availability

New DNA sequencing data has been deposited to GEO under accession code GSE183622. All data generated or analyzed are included in the manuscript and supplementary files. Source data files containing the numerical data for Figure 1 and Figure 2 are uploaded. Source codes used for sequencing reads mapping to identify alkylation lesions and repair analysis at yeast Abf1 and Reb1 binding sites are also uploaded.

The following dataset was generated:

| Author(s) | Year | Dataset title | Dataset URL | Database and Identifier |
|---|---|---|---|---|
| Duan M, Antony JS, Ulibarri J, Poon GMK, Wyrick JJ, Mao P, Sivapragasam S, Hinz JM | 2021 | Analysis of alkylation damage formation and base excision repair at yeast transcription factor binding sites | https://www.ncbi.nlm.nih.gov/geo/query/acc.cgi?acc=GSE183622 | NCBI Gene Expression Omnibus, GSE183622 |

The following previously published datasets were used:

| Author(s) | Year | Dataset title | Dataset URL | Database and Identifier |
|---|---|---|---|---|
| Duan M, Selvama K, Wyrick JJ, Mao P | 2020 | CPD-seq mapping of transcription-coupled DNA repair in yeast | https://www.ncbi.nlm.nih.gov/geo/query/acc.cgi?acc=GSE145911 | NCBI Gene Expression Omnibus, GSE145911 |
| Mao P, Wyrick JJ | 2017 | Genome-wide Maps of Alkylation Damage, Repair, and Mutagenesis in Yeast Reveal Mechanisms of Mutational Heterogeneity | https://www.ncbi.nlm.nih.gov/geo/query/acc.cgi?acci=GSE98031 | NCBI Gene Expression Omnibus, GSE98031 |
| Rossi MJ, Franklin Pugh B | 2021 | A high-resolution protein architecture of the budding yeast genome | https://www.ncbi.nlm.nih.gov/geo/query/acc.cgi?acc=GSE147927 | NCBI Gene Expression Omnibus, GSE147927 |

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

# Appendix 1

## Supplemental materials and Methods

### NMP-seq data reproducibility and statistical tests

The data reproducibility between two independent NMP-seq experiments was tested by Pearson correlation coefficient analysis. 7meG reads were counted at each position from –500 to +500 bp relative to the midpoint of the TF binding site and normalized to the number of total mapped reads. The 7meG density for Abf1 and Reb1 binding sites (i.e. 7meG per million reads) from two NMP-seq repeats was plotted for each yeast strain. Graphs and Pearson coefficients (Pearson's r) are shown in *Figure 1—figure supplement 3*.

To test the statistical significance of damage difference between TF binding sites and the flanking DNA in *Figure 1*, we randomly binned TF binding sites to five bins, with each bin containing ~ 100 binding sites. The mean and standard deviation of damage levels were analyzed among the five bins. The statistical data for 5 bp moving windows and 1 bp high-resolution analysis were shown in *Figure 1—figure supplement 2*.

### Validation of BER at TF binding sites

The BER analyses shown in *Figure 2* were performed using TF binding data generated with occupied regions of genomes from affinity-purified naturally isolated chromatin (ORGANIC), a method utilizing micrococcal nuclease (MNase) to digest native chromatin (i.e. not formaldehyde cross linked) and immunoprecipitate the TF-DNA complex for sequencing (*Kasinathan et al., 2014*). To validate the findings in *Figure 2*, we used TF binding data generated with the ChIP-exonuclease (ChIP-exo) method (*Rossi et al., 2021*). The TF ChIP-exo data were downloaded from the Gene Expression Omnibus, https://www.ncbi.nlm.nih.gov/geo/ (accession number GSE147927). ChIP-exo is similar to the conventional ChIP-seq, but utilizes exonuclease to cleave free DNA after chromatin immunoprecipitation to improve mapping resolution (*Rhee and Pugh, 2012*). Analysis of NMP-seq data at Abf1 and Reb1 ChIP-exo peaks and flanking regions showed strongly inhibited BER after 2 h repair (*Figure 2—figure supplement 3*, left and middle panels), consistent with analysis in *Figure 2* using the ORGANIC binding data. Moreover, ChIP-exo was used to map binding sites for other yeast TFs such as Repressor Activator Protein (Rap1) (*Rossi et al., 2021*), an essential yeast TF involved in both activation and suppression of RNA Pol II transcription. We analyzed 7meG repair at Rap1 ChIP-exo sites and found that BER was also strongly inhibited by Rap1 binding (*Figure 2—figure supplement 3*, right panel). Hence, NMP-seq analysis using both ORGANIC and ChIP-exo binding data consistently indicates an inhibitory role of TF binding in BER.

