## [Editor Report]

This manuscript will be of interest to researchers interested in DNA repair and transcriptional regulation. The authors provide a series of well-executed and designed high-resolution sequencing data demonstrating that transcription factor (TF) binding perturbs alkylation base damage formation as well as inhibits its repair via base excision repair (BER) at TF binding sites. Moreover, they demonstrate differences between nucleotide excision repair and BER at TF binding sites that are consistent with the different repair mechanism of these two pathways. These results should have an important and timely impact on the field. The revision addresses all points of the review and has significantly strengthened the manuscript and presentation.

---

## [Decision Letter]

**Decision letter after peer review:**

Thank you for submitting your article "High-resolution mapping of DNA alkylation damage and base excision repair at yeast transcription factor binding sites" for consideration by *eLife*. Your article has been reviewed by 3 peer reviewers, and the evaluation has been overseen by a Reviewing Editor and Kevin Struhl as the Senior Editor. The following individuals involved in review of your submission have agreed to reveal their identity: Bennett Van Houten (Reviewer #1); Amy Whitaker (Reviewer #2).

Essential revisions:

The results presented in this study provide an important new information on the effects of two transcription factors on formation and repair of damage at hundreds of binding sites within the yeast genome. The manuscript is well-written, although the way some of the experimental results were split in the main text and supplemental figures made the overall manuscript a bit cumbersome to read through. The authors also missed an opportunity for a nice summary figure showing high resolution mapping of lesions and repair kinetics within the Reb1 binding sites.

1) Reproducibility and statistics. The entire manuscript seems to be absent of statistical analysis. Please indicate the number of times each experiment was performed and how reproducible these data are from experiment to experiment. Finally, even within an experiment when hundreds of binding sites are compared what is the mean and S.D. around each point, what is the significance of 40% and 70% changes over a 5 bp window and a 1.5 increase at specific bases. What is the most appropriate statistical test to assess these differences? This type of deep dive into one data set is essential (perhaps described in a supplemental data section) to understand whether differences in repair which are presented are truly significantly different.

2) All reviewers recommend significant reorganization of the manuscript along the lines listed below:

– The validation studies using ORGANIC and Chip-Exo would be better shifted into the supplemental materials.

– Repair kinetics: The authors state on line 157 repair incubation (e.g. 1 and 2 h repair), but almost every figure legends in the main text indicates a 2 hr recovery periods were used to compare, however 1 hr data is reported in the supplement. Why break up the data in this way, and not show the kinetics of repair from 0, 1, and 2 all on the same graph or try to actually get a rate constants for removal based on these three time points? Also high resolution frequency initial damage is given, but then repair kinetics are often compared over broader regions.

Admittedly it is beyond the scope of this already demanding set of experiments to get high accuracy on precise repair kinetic profiles, but their subsequent analysis of repair kinetics was grouped by first looking at nucleosome-depleted regions and then suggesting that the slower repair within these regions are within the Abf1 and Reb1 binding position. Why did the authors change the comparison between Figure 1 and Figure 4? Could not individual high resolution repair kinetics be followed in a similar manner as the frequency distributions at base pair resolution? Then in Figure 5 the authors seems to us the high resolution matrix approach for the two TF binding sites to examine repair for 3meA. Why put the work-up of the +4 position of Reb1 high occupancy which showed slower kinetics in the supplement?

– As part of the enhancing the Discussion section, where specific structural details of the TF binding interactions are reviewed, it would be extremely informative and helpful to the high-resolution base-pair studies within the Reb1 binding sites. As long as these differences are statistically valid and real then showing the high-resolution damage mapping AND repair kinetics within the binding site from a co-crystal structure in a final summary figure would be elegant and helpful. For example, the structure shown in S3 C and D showing both a plot of damage frequencies and rates of removal would be extremely informative.

– Because of data presented Figure 6 regarding the inhibition of repair of UV-induced damage flanking the TF binding sites, the manuscript title is too limited and undermines the broad significance of this work and a more inclusive title discussing both types of damage and repair pathways could be included, for example, something like, "High resolution mapping demonstrate inhibition of DNA excision repair pathways by transcription factors". Clearly the effect of TF repair inhibition of UV-induced damage should be mentioned in the abstract as well.

– Improve clarity throughout the manuscript which data sets were used to determine binding sites and TF binding, and which for damage site analysis.

3) Figure 1 – Analysis of alkylated DNA bases immediately after the MMS exposure should be performed also in WT cells both to strengthen the manuscript that further primarily uses WT strain as well as to achieve adequate normalization to naked DNA isolated from WT cells. This requires additional experimentation.

4) Figure 2C (pg 8, line 166-167) – Authors indicate with respect to the high and low occupancy of Reb1 sites that "The repair suppression is mediated by TF binding, not the underlying DNA sequence…". However, at those sites that nucleosome occupancy appears also to be markedly different and thus could potentially impact repair efficiency. This, as well as in general the combined effects of both TF binding and accompanying chromatin organization (e.g. Figure 3A and 3C), should be further discussed.

5) Figure 4 – While the results indicate clear specific accumulation of alkylated bases in the promoter regions of Abf1 and Reb1 target genes, the impact of impaired repair on the expression of these genes should be tested. Specifically, since recent works suggested that AAG-BER and BER in general can influence gene expression. The author may want to consider additional experimentation and/or discussing this limitation of their data.

6) Figure 5E – BSA should be used as a negative control in reactions without Reb1 to make sure that the effect is not driven simply by increased overall protein concentration. This could be combined with the effort to present results of at least three independent experiments in Figure 5F. This requires additional experimentation.

7) Figure 4 and 5 – Analysis of Abf1 and Reb1 in promoters of certain target genes should be analyzed directly and after the MMS treatment.

8) Figure S7C – Result should be quantified.

Additional points

These require text changes only.

9) pg. 14 – While inosine is a stable AAG substrate, it is not a direct analog of 3meA. This should be corrected.

10) Authors are encouraged to cite more the original works instead of review articles e.g. pg.3 "During BER, AAG/Mag1 removes the alkylated base and generates an AP site, which is then cleaved by the apurinic/apyrimidinic endonuclease (APE1) (Whitaker and Freudenthal, 2018)."

11) Figure 3: Can the authors explain why the regions more distant from the TF binding sites (i.e. -500 to -200 and 200 to 500) have reduced repair in the Reb1-AA and Abf1-AA cells compared to the WT-AA cells?

12) Figure S6 – indicate in the figure legend which data used for Abf1 and Reb1 binding.

13) The authors compare differences between BER and NER mechanisms that support their finding that TF binding inhibits a larger region of DNA for NER. One point worth considering adding is that BER factors only form transient interactions with one another, as opposed to NER which is thought to form a more stable multi-protein complex on the DNA and thus would likely have a larger overall DNA footprint during repair.

---

## [Author Response]

1) Reproducibility and statistics. The entire manuscript seems to be absent of statistical analysis. Please indicate the number of times each experiment was performed and how reproducible these data are from experiment to experiment. Finally, even within an experiment when hundreds of binding sites are compared what is the mean and S.D. around each point, what is the significance of 40% and 70% changes over a 5 bp window and a 1.5 increase at specific bases. What is the most appropriate statistical test to assess these differences? This type of deep dive into one data set is essential (perhaps described in a supplemental data section) to understand whether differences in repair which are presented are truly significantly different.

We thank the reviewers for this important suggestion. Each of the NMP-seq experiments (e.g., *mag1∆*, WT, anchor-away) was performed twice independently and the information is now added to Materials and methods (page 23). Data reproducibility has been tested using Pearson correlation coefficient analysis. The data shows high reproducibility between two independent NMPseq experiments (e.g., Pearson’s r > 0.8). The data is now added to Supplemental figure 3 (note: the full name is Figure 1— figure supplement 3).

We have now conducted statistical tests to assess difference between the TF binding motif and flanking regions within the same experiment for Figure 1, as suggested by the reviewers. Considering there are hundreds of binding sites for Abf1 and Reb1, we adapted the concept of a statistical method named Bootstrap (Efron, 1979) and randomly binned Abf1 or Reb1 binding sites into five equal-sized bins, with each bin containing ~100 binding sites. We then analyzed mean and S.D. of 7meG reads from the five bins. The statistical significance was analyzed by paired t tests of 5-bp moving windows as well as at each single nucleotide. These analyses showed significantly changed 7meG within the binding motif relative to the flanking DNA with p values < 0.05. We have now added sentences in the Results section to mention that the differences are statistically significant (page 6). The statistical test method is described in Supplemental Materials and methods and the data is added to Supplemental figure 2.

2) All reviewers recommend significant reorganization of the manuscript along the lines listed below:– The validation studies using ORGANIC and Chip-Exo would be better shifted into the supplemental materials.

We agree. We have moved the validation from the main text to the Supplemental Materials and methods.

– Repair kinetics: The authors state on line 157 repair incubation (e.g. 1 and 2 h repair), but almost every figure legends in the main text indicates a 2 hr recovery periods were used to compare, however 1 hr data is reported in the supplement. Why break up the data in this way, and not show the kinetics of repair from 0, 1, and 2 all on the same graph or try to actually get a rate constants for removal based on these three time points? Also high resolution frequency initial damage is given, but then repair kinetics are often compared over broader regions.Admittedly it is beyond the scope of this already demanding set of experiments to get high accuracy on precise repair kinetic profiles, but their subsequent analysis of repair kinetics was grouped by first looking at nucleosome-depleted regions and then suggesting that the slower repair within these regions are within the Abf1 and Reb1 binding position. Why did the authors change the comparison between Figure 1 and Figure 4? Could not individual high resolution repair kinetics be followed in a similar manner as the frequency distributions at base pair resolution? Then in Figure 5 the authors seems to us the high resolution matrix approach for the two TF binding sites to examine repair for 3meA. Why put the work-up of the +4 position of Reb1 high occupancy which showed slower kinetics in the supplement?

We conducted repair experiments for both 1 h and 2 h and the repair pattern was similar between the two time points. We only presented the 2 h data in the main text because it shows stronger repair inhibition at TF binding sites and in the flanking nucleosomes.

We followed the suggestion and plotted 0, 1, and 2 h data in the same graph (see Author response image 1), but the figure looks overcrowded and it is hard to see damage peaks at the central motif for 1 and 2 h. Hence, we chose to present 1 and 2 h data in two separate graphs in a new Figure 2 to show the repair trend. The single base resolution repair data (1 h and 2 h) has also been added to the new Figure 2, similar to the plot showing high-resolution initial damage in Figure 1. The data is explained in the Results section on page 9.

**Author response image 1. sa2fig1:** Damage levels at 0, 1, and 2 hr for Abf1 binding sites plotted in the same graph.

The analysis in Figure 4 is slightly different from Figure 1 (and 2) in that genes were aligned at the transcription start site (TSS) to focus on repair in the context of transcription. This allows us to highlight repair inhibition by TF binding in gene promoters and show the inhibited location relative to the TSS. The purpose of Figure 1 and 2 is to compare damage/repair between the motif sequence and the flanking nucleosomal DNA. Therefore, we used the mapped TF binding sites for Figure 1 and 2, but the annotated gene TSS for Figure 4.

The slow repair of 3meA at the +4 position of Reb1 binding sites was shown in the supplemental data to explain the rationale of the in vitro experiments in Figure 5. It helps the readers understand why we designed inosine damage at the +4 position.

– As part of the enhancing the Discussion section, where specific structural details of the TF binding interactions are reviewed, it would be extremely informative and helpful to the high-resolution base-pair studies within the Reb1 binding sites. As long as these differences are statistically valid and real then showing the high-resolution damage mapping AND repair kinetics within the binding site from a co-crystal structure in a final summary figure would be elegant and helpful. For example, the structure shown in S3 C and D showing both a plot of damage frequencies and rates of removal would be extremely informative.

We appreciate this insightful suggestion and have now generated a new supplemental figure (Figure 2 — figure supplement 2), which includes the Reb1-DNA co-crystal structure, highresolution damage frequency, and high-resolution repair data at 1 and 2 h. The content of this figure is discussed in detail in the Discussion section on page 19-20.

– Because of data presented Figure 6 regarding the inhibition of repair of UV-induced damage flanking the TF binding sites, the manuscript title is too limited and undermines the broad significance of this work and a more inclusive title discussing both types of damage and repair pathways could be included, for example, something like, "High resolution mapping demonstrate inhibition of DNA excision repair pathways by transcription factors". Clearly the effect of TF repair inhibition of UV-induced damage should be mentioned in the abstract as well.

This is a good suggestion and will broaden the scope of this study. We have changed the title to “High-resolution mapping demonstrates inhibition of DNA excision repair by transcription factors”. Repair inhibition of UV damage is mentioned in the abstract.

– Improve clarity throughout the manuscript which data sets were used to determine binding sites and TF binding, and which for damage site analysis.

To clarify, we used ORGNIC TF binding sites for most of our analysis, except for the ChIP-exo validation, which used the ChIP-exo binding sites. We have now moved the ChIP-exo part to Supplemental Materials and methods to reduce the confusion. Additionally, we revised the Materials and methods section in the main manuscript to clarify that all analysis was conducted using the ORGANIC data.

For NMP-seq datasets, we used data generated in different yeast strains, including WT, *mag1* mutant, and anchor-aways. We have explained which strain was used in the Results section.

3) Figure 1 – Analysis of alkylated DNA bases immediately after the MMS exposure should be performed also in WT cells both to strengthen the manuscript that further primarily uses WT strain as well as to achieve adequate normalization to naked DNA isolated from WT cells. This requires additional experimentation.

The WT-0 h library was sequenced in parallel with the WT-1 h and 2 h repair time points, but was not used in the original version of the manuscript. We have now revisited the WT-0 h data and generated a new supplemental figure (Figure 1 — figure supplement 5). Analysis of WT-0 h data (normalized to naked DNA) shows a very similar damage formation pattern as in the *mag1*-0 h, suggesting the endogenous repair does not remove a significant portion of damage during the 10minute MMS treatment time. We have revised the Results section on page 7, to mention the WT-0 h data.

4) Figure 2C (pg 8, line 166-167) – Authors indicate with respect to the high and low occupancy of Reb1 sites that "The repair suppression is mediated by TF binding, not the underlying DNA sequence…". However, at those sites that nucleosome occupancy appears also to be markedly different and thus could potentially impact repair efficiency. This, as well as in general the combined effects of both TF binding and accompanying chromatin organization (e.g. Figure 3A and 3C), should be further discussed.

This comment raises a good point that the combination of TF binding and chromatin organization modulates BER. We have expanded the discussion on how the chromatin organization affects repair at high and low Reb1 sites (page 9) and in the anchor-away yeast strains (page 12) in the revised manuscript.

5) Figure 4 – While the results indicate clear specific accumulation of alkylated bases in the promoter regions of Abf1 and Reb1 target genes, the impact of impaired repair on the expression of these genes should be tested. Specifically, since recent works suggested that AAG-BER and BER in general can influence gene expression. The author may want to consider additional experimentation and/or discussing this limitation of their data.

We have followed this suggestion to expand the Discussion (page 20). This expansion discusses potential effects of the unrepaired alkylation damage, mutations caused by the unrepaired damage, and BER enzymes, on gene expression. While this manuscript is focused on damage and repair and does not directly address gene expression, we are interested in investigating this important topic in a future study.

6) Figure 5E – BSA should be used as a negative control in reactions without Reb1 to make sure that the effect is not driven simply by increased overall protein concentration. This could be combined with the effort to present results of at least three independent experiments in Figure 5F. This requires additional experimentation.

All repair reactions, including naked DNA without Reb1, were done in the presence of 100 µg/ml BSA. This is now communicated more clearly in the methods section relevant to our in vitro experiments (page 27).

We purified new Reb1 protein and performed the in vitro repair experiments three more times. The quantification data from four independent repeats, showing the average repair rates and S.D., is now included in a new Figure 5F. All the original unedited repair gel images are also submitted.

7) Figure 4 and 5 – Analysis of Abf1 and Reb1 in promoters of certain target genes should be analyzed directly and after the MMS treatment.

We consulted with the reviewing editor and confirmed that this comment is about DNA damage/repair at Abf1 and Reb1 binding sites in promoters of individual target genes. We have added new gene-by-gene heatmaps to demonstrate 7meG damage in each target gene in a supplemental figure (Figure 4 —figure supplement 1), which showed correlation between BER inhibition and TF binding.

8) Figure S7C – Result should be quantified.

We have now quantified the gels and the quantification data is added to the figure.

Additional pointsThese require text changes only.9) pg. 14 – While inosine is a stable AAG substrate, it is not a direct analog of 3meA. This should be corrected.

We corrected it in the main text on page 15.

10) Authors are encouraged to cite more the original works instead of review articles e.g. pg.3 "During BER, AAG/Mag1 removes the alkylated base and generates an AP site, which is then cleaved by the apurinic/apyrimidinic endonuclease (APE1) (Whitaker and Freudenthal, 2018)."

We appreciate this reminder and have updated the citation.

11) Figure 3: Can the authors explain why the regions more distant from the TF binding sites (i.e. -500 to -200 and 200 to 500) have reduced repair in the Reb1-AA and Abf1-AA cells compared to the WT-AA cells?

We think two mechanisms can potentially contribute to the reduced repair in Reb1-AA and Abf1-AA cells. First, both Reb1 and Abf1 are general regulatory factors (GRFs) and regulate expression of many yeast genes (Chasman et al., 1990; Miyake et al., 2004). Depletion of them from the nucleus likely affects gene expression, including DNA repair genes. Second, it has been shown that Abf1 and Reb1 cooperate with the chromatin remodeling complex, RSC, to establish accessible chromatin to facilitate transcription (Kubik et al., 2018). The less accessible chromatin and disrupted chromatin organization in the anchor-away cells may reduce repair efficiency considerably.

12) Figure S6 – indicate in the figure legend which data used for Abf1 and Reb1 binding.

We have now indicated in the figure legend that the NMP-seq data generated in WT-2 h cells was used. Repair was analyzed in genes linked with Abf1 and Reb1 binding sites mapped by the ORGANIC method (Kasinathan et al., 2014).

13) The authors compare differences between BER and NER mechanisms that support their finding that TF binding inhibits a larger region of DNA for NER. One point worth considering adding is that BER factors only form transient interactions with one another, as opposed to NER which is thought to form a more stable multi-protein complex on the DNA and thus would likely have a larger overall DNA footprint during repair.

We have mentioned the more stable NER complex on DNA as an additional mechanism for the larger repair inhibition region in Discussion (page 21).

References:

Chasman DI, Lue NF, Buchman AR, LaPointe JW, Lorch Y, Kornberg RD. 1990. A yeast protein that influences the chromatin structure of UASG and functions as a powerful auxiliary gene activator. Genes Dev 4:503–514. doi:10.1101/gad.4.4.503

Efron B. 1979. Bootstrap Methods: Another Look at the Jackknife. The Annals of Statistics 7:1–26. doi:10.1214/aos/1176344552

Kasinathan S, Orsi GA, Zentner GE, Ahmad K, Henikoff S. 2014. High-resolution mapping of transcription factor binding sites on native chromatin. Nat Methods 11:203–209.

doi:10.1038/nmeth.2766

Kubik S, O’Duibhir E, de Jonge WJ, Mattarocci S, Albert B, Falcone J-L, Bruzzone MJ, Holstege FCP, Shore D. 2018. Sequence-Directed Action of RSC Remodeler and General Regulatory Factors Modulates +1 Nucleosome Position to Facilitate Transcription. Molecular Cell 71:89-102.e5. doi:10.1016/j.molcel.2018.05.030

Miyake T, Reese J, Loch CM, Auble DT, Li R. 2004. Genome-wide analysis of ARS (autonomously replicating sequence) binding factor 1 (Abf1p)-mediated transcriptional regulation in *Saccharomyces cerevisiae*. J Biol Chem 279:34865–34872. doi:10.1074/jbc.M405156200